# Depression, Anxiety, and Health-Related Quality of Life in Adults with Rheumatoid Arthritis: Findings from a National Survey

**DOI:** 10.3390/jcm14227940

**Published:** 2025-11-09

**Authors:** Monira Alwhaibi

**Affiliations:** Department of Clinical Pharmacy, College of Pharmacy, King Saud University, Riyadh 11451, Saudi Arabia; malwhaibi@ksu.edu.sa

**Keywords:** rheumatoid arthritis, health-related quality of life, depression, anxiety, MEPS, mental health, physical activity

## Abstract

**Background**: Rheumatoid arthritis (RA) is a chronic autoimmune disease that substantially impairs health-related quality of life (HRQoL). Comorbid mental health conditions, particularly depression and anxiety, may further exacerbate this burden, yet evidence from large, population-based studies remains limited. Therefore, this study examined the association between comorbid depression and anxiety and HRQoL among adults with RA using nationally representative data from the United States. **Methods**: Data were drawn from the 2017–2022 Medical Expenditure Panel Survey. Adults aged ≥18 years with self-reported RA were included. HRQoL was assessed using the Veterans RAND 12-Item Health Survey (VR-12) physical (PCS) and mental (MCS) component summary scores. Multiple linear regression models were used to evaluate associations between depression, anxiety, and HRQoL, adjusting for sociodemographic, behavioral, and health-related covariates. **Results**: Comorbid depression and anxiety were significantly associated with lower HRQoL scores compared with RA alone. Participants with both conditions exhibited the poorest PCS and MCS scores, indicating a disease burden. Lower income, unemployment, and limited physical activity were also linked to poorer HRQoL, whereas better self-rated health and physical activity were positive predictors. **Conclusions**: Depression and anxiety independently and jointly contribute to poorer HRQoL among adults with RA, even after controlling for key confounders. These findings highlight the importance of integrated care models that address both psychological and physical health, alongside interventions promoting physical activity to enhance overall well-being.

## 1. Introduction

Rheumatoid arthritis (RA) is a chronic, systemic autoimmune disease that primarily affects the joints, leading to inflammation, pain, stiffness, and progressive disability [1]. It has also been defined as synovitis in at least one joint with no better alternative diagnosis and a scoring ≥ 6 (out of 10) across involved joints, serologic markers, acute-phase response and symptom duration, under the ICD-11-CM codes [2]. RA affects approximately 0.5% to 1% of the global population and occurs more frequently in women than in men [3,4]. RA represents a major cause of morbidity among adults worldwide, resulting in substantial physical, psychological, and social burden throughout the life course [5]. The chronic pain and fatigue characteristic of RA, combined with its unpredictable flares and potential for irreversible joint damage, can severely impair functional capacity and emotional well-being [6]. Moreover, RA contributes significantly to humanistic and economic burden through reduced productivity, work disability, diminished quality of life, and increased healthcare costs and utilization [7,8].

Beyond physical disability, RA is strongly associated with psychological comorbidities, particularly depression and anxiety, and the links between depression and RA appear to be bidirectional [9]. Previous studies have demonstrated that between 13% and 47% of individuals with RA experience depressive symptoms [10]. These mental health conditions not only exacerbate physical symptoms but also influence treatment adherence, perception of pain, and overall disease progression [10]. Depression and anxiety are also linked to disability, reduced work productivity, higher healthcare utilization and costs, and increased mortality among RA patients [11,12,13,14,15]. Consequently, assessing HRQoL in the context of mental health comorbidities is vital to understanding the full burden of RA.

Health-related quality of life (HRQoL) refers to “how well a person functions in their life and his or her perceived well-being in physical, mental, and social domains of health” [16]. In rheumatoid arthritis, HRQoL is a key outcome measure that reflects the broader impact of the disease beyond clinical symptoms or laboratory markers. Previous research has consistently demonstrated that poor mental health, particularly depression and anxiety, is strongly associated with reduced HRQoL among individuals with RA [15,17,18,19,20]. Multiple interrelated factors contribute to impaired HRQoL in this population, including chronic pain, disease severity, fatigue, and functional disability, all of which can exacerbate psychological distress and diminish overall well-being [20,21,22,23]. Furthermore, lower HRQoL has been linked to increased healthcare utilization, higher medical costs, and greater dependence on health services [24].

Despite growing recognition of the importance of HRQoL in RA management, previous studies examining its determinants have been limited by small, non-representative samples, cross-sectional designs, and a lack of control for key confounders. Moreover, most investigations have focused on depression alone, neglecting the frequent coexistence and interactive effects of depression and anxiety on HRQoL [15,17,18,19,20]. The present study aims to address these gaps by utilizing data from the 2017–2022 Medical Expenditure Panel Survey (MEPS), a nationally representative dataset of the U.S. noninstitutionalized population. By employing a large, diverse sample and controlling for multiple sociodemographic, behavioral, and clinical covariates known to influence HRQoL, this study provides a more rigorous and generalizable assessment of the relationship between comorbid depression and anxiety and HRQoL in adults with RA. Generating nationally representative evidence on these interrelationships will enhance understanding of the mental–physical health interface in RA.

## 2. Methods

### 2.1. Study Design and Data

This study used a retrospective longitudinal design with data drawn from the 2017–2022 cycle of the Medical Expenditure Panel Survey (MEPS), a national survey of the U.S. population administered by the Agency for Healthcare Research and Quality. MEPS collects information from households across two consecutive years through a stratified, multistage probability sampling framework. Data are gathered in multiple rounds using computer-assisted personal interviews. MEPS provides comprehensive data on demographic and socioeconomic characteristics, chronic health conditions, healthcare use and expenditures, health insurance, income, and employment. Medical conditions including depression and anxiety are self-reported by participants during interviews and then coded into the International Classification of Diseases Clinical Modification (ICD-10-CM) codes, the tenth revision, by trained professional coders.

### 2.2. Study Population

Participants were eligible for inclusion if they were 18 years old and above, had a documented diagnosis of rheumatoid arthritis, were alive during the study period, and had complete HRQoL data. RA cases were identified in the MEPS dataset using the ICD-10-CM code “M06” (other RA) [25].

### 2.3. Measures

#### 2.3.1. Study Outcome: Health-Related Quality of Life

To assess health-related quality of life (HRQoL), participants in MEPS completed the Veterans RAND 12-Item Health Survey (VR-12©), a widely used and validated patient-reported outcome measure [26,27,28]. The VR-12 captures eight dimensions of health: general health, physical functioning, role limitations due to physical health, bodily pain, vitality, role limitations due to emotional problems, mental health, and social functioning [23]. From these domains, two composite scores are generated: the Physical Component Summary (PCS), which reflects general health, physical functioning, role physical, and bodily pain; and the Mental Component Summary (MCS), which reflects vitality, role emotional, mental health, and social functioning [29].

#### 2.3.2. Independent Variables

The primary independent variable in this study was rheumatoid arthritis (RA) groups, classified into four mutually exclusive groups: RA only, RA with anxiety, RA with depression, and RA with both anxiety and depression. Other independent variables were included based on the existing evidence of their impact on HRQoL in adults with RA. These encompassed sociodemographic factors (age group, gender, education level, household income, employment status, marital status, and region of residence). Health insurance was included as a proxy for healthcare access, which may directly influence HRQoL outcomes. Health behavior, such as physical activity, and self-rated health were also considered. In addition, chronic comorbidities (e.g., hypertension, diabetes, asthma, COPD, GERD) were accounted for, as they are highly prevalent among individuals with RA and contribute to overall disease burden and diminished quality of life.

### 2.4. Statistical Analyses

Descriptive statistics were used to summarize the baseline characteristics of RA sample, including frequencies, and percentages. Differences in baseline characteristics across the RA groups were evaluated using bivariate analysis, (chi-square tests). Variations in mean HRQoL scores by RA groups were examined using ANOVA, followed by Tukey’s post hoc tests to determine which specific groups differed significantly. To evaluate the association between RA groups and HRQoL, multivariable linear regression model was conducted, adjusting for all relevant covariates. All estimations accounted for the complex survey design of MEPS, incorporating variance adjustment weights (strata and primary sampling unit) along with person-level weights. Statistical analyses were performed using SAS 9.4 (SAS Institute Inc., Cary, NC, USA).

## 3. Results

### 3.1. Characteristics of the Rheumatoid Arthritis Sample

Table 1 presents the characteristics of the study sample (n = 2324), and stratified by RA with and without comorbid depression and anxiety. Overall, 69.6% of participants had RA only, while 11.8% had RA with depression, 10.7% had RA with anxiety, and 7.9% had RA with both depression and anxiety. Age was significantly associated with RA groups (*p* < 0.001), with younger adults (22–39 years) more likely to present with depression and/or anxiety, compared to older adults (>64 years). Women were overrepresented in the RA with depression and/or anxiety groups compared with men (*p* < 0.001). Regional variation was observed (*p* = 0.013), with participants from the Northeast more likely to have RA with depression and/or anxiety compared to other regions. Socioeconomic characteristics were also associated with RA groups. Employment status (*p* = 0.021) and poverty status (*p* = 0.022) showed significant relationships, as unemployed and poor participants were more likely to have RA with depression and/or anxiety compared to those employed or with high income. Insurance coverage was also important: participants with public insurance had higher rates of RA with depression/anxiety compared to those private insurance (*p* = 0.013). Poorer general health was strongly associated with comorbid depression and/or anxiety (*p* < 0.001); nearly 40% of those reporting fair or poor health were in these groups. Similarly, physical activity was significant (*p* = 0.001), with those not exercising more likely to have RA with depression. Several comorbid conditions were significantly associated with RA with depression and/or anxiety. These included asthma (*p* = 0.001), chronic obstructive pulmonary disease (*p* < 0.001), osteoarthritis (*p* = 0.002), and gastroesophageal reflux disease (*p* < 0.001).

Overall, effect size estimates indicated that most associations between rheumatoid arthritis group status and demographic, socioeconomic, and health-related characteristics were of small to moderate magnitude, suggesting meaningful but not large differences across groups (Appendix A).

### 3.2. Health-Related Quality of Life and Rheumatoid Arthritis Groups

Table 2 displays the weighted means and standard errors of HRQoL scores by RA groups. Both the physical component summary (PCS) and mental component summary (MCS) scores differed significantly across groups (*p* < 0.0001). Participants with RA only had the highest mean PCS (38.53, SE = 0.46) compared to those with RA and comorbid conditions, while those with RA, depression, and anxiety had the lowest mean PCS (33.99, SE = 1.26). A similar gradient was observed for MCS. The RA-only group reported the highest MCS score (51.15, SE = 0.33), followed by those with RA and anxiety (43.85, SE = 0.92), RA and depression (42.74, SE = 1.05), and the lowest among participants with RA, depression, and anxiety (38.55, SE = 1.01).

The one-way ANOVA effect size results indicate that RA groups had a significant effect on both PCS and MCS (Appendix A). The effect of RA group on PCS was statistically significant (F(3, 2320) = 15.79, *p* < 0.0001), but the effect size was small (partial η^2^ = 0.0199; ω^2^ = 0.0189). This suggests that while there are measurable differences in physical health across the groups, RA comorbidity status explains only about 2% of the variance in PCS scores. The effect of RA group on MCS was highly significant (F(3, 2320) = 164.22, *p* < 0.0001), with a large effect size (partial η^2^ = 0.175; ω^2^ = 0.175). This indicates that RA comorbidity status explains approximately 17.5% of the variance in mental QoL scores, reflecting a substantial impact of depression and/or anxiety on mental well-being among RA patient conditions (Appendix A). Post hoc results indicate that comorbid depression and anxiety were associated with substantial reductions in both PCS and MCS among individuals with RA, with the greatest decline observed in those reporting both conditions (Appendix A).

### 3.3. Health-Related Quality of Life Among Rheumatoid Arthritis Groups from Linear Regression Analysis

Table 3 presents the adjusted multivariable linear regression estimates for predictors of HRQoL among adults with RA. Significant differences were observed across RA subgroups. Compared to participants with RA only, those with RA and depression, RA and anxiety, and RA with both depression and anxiety had significantly lower MCS scores (B = −6.61, −6.84, and −9.77, respectively; all *p* < 0.001). For the PCS, RA with depression (B = −0.45, *p* < 0.001) and RA with anxiety (B = −1.10, *p* < 0.001) were associated with lower scores, whereas RA with both depression and anxiety did not significantly differ from RA only. Figure 1 illustrates the forest plot of adjusted regression coefficients of RA groups on HRQoL.

Age, gender, race/ethnicity, and marital status were also significant predictors of HRQoL. Younger adults (22–39 years) had higher PCS scores but lower MCS scores compared with older adults (>65 years) (*p* < 0.001). Men reported slightly better PCS (B = 0.20, *p* < 0.05) and MCS (B = 0.09, *p* < 0.001) scores compared with women. African American, Latino, and “other” race groups all showed higher PCS scores relative to Whites, but MCS scores varied, with Latinos (B = −0.67, *p* < 0.001) and “others” (B = −2.04, *p* < 0.001) reporting significantly lower MCS. Married participants reported better MCS scores compared with never-married adults.

Socioeconomic status and access to resources were strongly associated with HRQoL. Employment was linked with substantially higher PCS (B = 4.44, *p* < 0.001) and MCS (B = 0.94, *p* < 0.001). Compared to high-income participants, those who were poor, near-poor, or middle-income reported significantly lower PCS and MCS scores (all *p* < 0.001). Prescription drug insurance was associated with lower PCS (B = −1.32, *p* < 0.001) and MCS (B = −2.14, *p* < 0.001). Regional differences were also noted: residents of the Midwest had higher PCS and MCS scores compared with those in the West, while those in the South reported lower PCS.

Health and lifestyle factors contributed strongly to HRQoL. Better self-rated general health was associated with higher PCS and MCS, with excellent/very good health yielding the largest positive associations (PCS: B = 12.32; MCS: B = 5.12; *p* < 0.001). Engaging in physical activity at least three times per week was associated with better PCS (B = 3.44, *p* < 0.001) and MCS (B = 1.25, *p* < 0.001). Several comorbidities were linked to lower HRQoL. Heart disease, hypertension, diabetes, asthma, COPD, and GERD all negatively predicted PCS. Similarly, diabetes, asthma, GERD, and cancer were linked with lower MCS. Effect sized results are shown in (Appendix A), the regression analyses revealed that depression and anxiety remained significant predictors of lower physical and mental HRQoL even after adjusting for sociodemographic and clinical covariates. The overall model fit corresponded to large effect sizes (Cohen’s f^2^ = 0.70 for the PCS and f^2^ = 0.46 for the MCS), suggesting that the predictors accounted for a meaningful proportion of variability in HRQoL outcomes. These results underscore the substantial practical impact of psychological comorbidities on perceived health among adults with rheumatoid arthritis.

## 4. Discussion

This study extends previous research by quantifying the magnitude and clinical relevance of comorbid depression and anxiety on HRQoL among adults with rheumatoid arthritis (RA) using nationally representative data. While prior studies have demonstrated that these mental health conditions worsen HRQoL, the present analysis is among the first to incorporate formal effect size measures, providing insight into the clinical significance of these associations beyond statistical significance alone. The results indicate that comorbid depression and anxiety exert a large and meaningful impact on mental HRQoL, explaining approximately 17% of the variance in mental component scores, whereas their effect on physical functioning is smaller but still significant. The large effect sizes observed in the multivariate models further demonstrate that psychosocial, demographic, and health-related variables collectively account for a substantial proportion of HRQoL variability. By jointly examining depression and anxiety rather than treating them as isolated factors, this study highlights the compounding burden of mental health comorbidities, particularly when coupled with socioeconomic disadvantage and limited physical activity, underscoring the multifactorial nature of well-being among individuals with RA.

Our results align with prior research conducted in diverse settings, which consistently show that psychiatric comorbidities impart the quality of life in RA [15,17,18,19,20]. Our findings corroborate this evidence and extend it to a broader U.S. adult population, highlighting the population-level implications of comorbid mental illness in RA. Importantly, our results highlight the compounded effect of coexisting depression and anxiety. Participants with both conditions exhibited the lowest HRQoL scores particularly in the mental health domain, suggesting a synergistic rather than additive burden. This observation aligns with previous evidence that depression and anxiety often co-occur in chronic illnesses and interact to amplify symptom perception, fatigue, and functional limitations. Such comorbidity may also hinder treatment adherence and self-management, further compromising long-term outcomes.

Beyond mental health, this study reinforces the substantial influence of social and behavioral determinants on HRQoL. Lower income, unemployment, and public insurance coverage were all associated with poorer HRQoL, reflecting the socioeconomic disparities that shape both disease experience and access to care. These findings echo prior research indicating that patients with limited financial resources face greater functional impairment and reduced treatment satisfaction.

Consistent with a recent systematic review that examined 21 studies and identified 70 determinants of HRQoL in RA, our results emphasize the multifactorial nature of quality of life in this population [30]. That review classified determinants into sociodemographic, RA-related, comorbidity, behavioral, and psychosocial domains, and found that poorer physical function, and the presence of comorbidities were consistently associated with reduced HRQoL. Among behavioral factors, exercise and sleep were the only determinants significantly linked to better HRQoL, while anxiety and depression emerged as the most prominent psychosocial factors predicting poorer HRQoL. Our findings reinforce and extend this evidence, showing that comorbid depression and anxiety remain significant predictors of diminished HRQoL even after controlling for these broader determinants.

Importantly, the identification of exercise as a key behavioral determinant of better HRQoL is supported by growing interventional evidence. A comprehensive synthesis of systematic reviews and meta-analyses has demonstrated that aerobic and structured exercise training are both safe and effective for patients with rheumatoid arthritis, leading to significant improvements in functional capacity, aerobic fitness, pain reduction, fatigue, vitality, psychological well-being, and overall HRQoL [31,32]. These findings collectively suggest that promoting physical activity alongside psychological support may represent an integrated, patient-centered approach to improving both physical and mental health outcomes in RA.

### 4.1. Study Strengths and Limitations

The present study makes several important contributions to the existing literature. First, it is among the few analyses to jointly examine depression and anxiety in relation to HRQoL in RA using a large, nationally representative U.S. sample. Second, the use of a validated HRQoL instrument (VR-12) and rigorous adjustment for multiple confounders enhances the validity and generalizability of the findings. Third, by quantifying the differential impact of comorbid mental health conditions, this research provides actionable insights for clinicians and policymakers to prioritize integrated care approaches that address both physical and psychological dimensions of RA. In addition, the present study advances the field by quantifying and interpreting effect size measures, an important step toward enhancing the interpretability and clinical relevance of statistical findings. However, certain limitations should be acknowledged. MEPS lacked data on RA disease duration, severity, treatment regimen, and disease activity measures, all of which are important determinants of quality of life in RA patients. The absence of these variables may have introduced unmeasured confounding. In addition, self-reported HRQoL may also be subject to recall or reporting bias.

### 4.2. Clinical, Public Health, and Research Implications

From a clinical perspective, these results of high mental health prevalence and impact HRQoL in Ra population highlight the importance of integrating mental health assessment into standard RA care pathways. Routine screening and timely referral to psychological services could improve disease management and overall quality of life. On a public health level, programs that promote physical activity can help alleviate the humanistic burden of RA. Future research should explore longitudinal relationships between disease activity, psychological distress, and HRQoL. Additionally, intervention studies that measure changes in effect sizes over time can help determine the real-world clinical impact of integrating mental health support into RA treatment protocols.

## 5. Conclusions

This nationally representative study demonstrates that comorbid depression and anxiety are strongly associated with poorer HRQoL among adults with rheumatoid arthritis, even after adjusting for key sociodemographic and clinical factors. The combined burden of these mental health conditions highlights the need for integrated management approaches that address both physical and psychological aspects of care. Additionally, socioeconomic disadvantage and low physical activity further contribute to reduced well-being, emphasizing the role of social and behavioral determinants. Interventions focused on mental health support and lifestyle modification may improve overall quality of life and should be prioritized in comprehensive rheumatoid arthritis care.

## Figures and Tables

**Figure 1 jcm-14-07940-f001:**
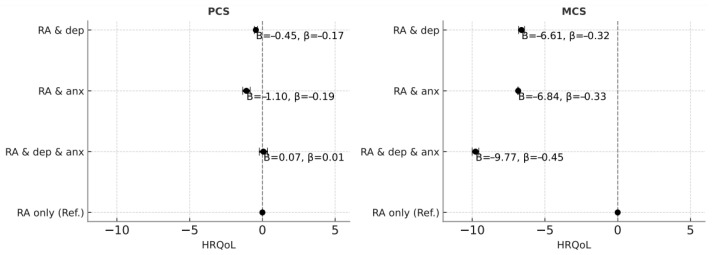
Forest plot of adjusted regression coefficients of RA groups on HRQoL.

**Table 1 jcm-14-07940-t001:** Description of the Study Sample (n = 2324), Number and weighted Row Percentage of Baseline Characteristics by Rheumatoid Arthritis Groups.

		Total Sample		RA Only		RA & Dep		RA & Anxiety		RA & Dep & Anxiety		
		N	Wt.%	N	Wt.%	N	Wt.%	N	Wt.%	N	Wt.%	*p*-Value
All		2324	100	1610	69.6	268	11.8	258	10.7	188	7.9	
Age in years												
	18–39	106	5.7	59	58.7	13	12.7	19	15.5	15	13.1	<0.0001
	40–49	190	8.6	112	57.8	23	11.6	24	13.3	31	17.3	
	50–64	861	38.4	573	68.2	119	14.3	91	8.9	78	8.6	
	>64	1167	47.3	866	74.2	113	9.8	124	11.0	64	5.0	
Gender												
	Women	1670	68.1	1091	65.6	209	12.7	213	12.5	157	9.2	<0.0001
	Men	654	31.9	519	78.2	59	10.0	45	6.8	31	5.1	
Race/ethnicity												
	White	1241	63.8	838	67.2	138	11.4	142	11.8	123	9.6	0.061
	African American	498	15.5	362	75.2	53	10.5	60	10.4	23	3.8	
	Latino	434	13.3	294	71.6	58	13.6	46	8.3	36	6.5	
	others	151	7.4	116	74.9	19	14.7	10	5.7	6	4.6	
Marital status												
	Married	1041	51.6	773	73.3	112	11.4	91	8.9	65	6.4	0.100
	Widow/Sep/Divorce	1016	39.3	667	66.8	125	11.5	132	12.3	92	9.4	
	Never married	267	9.2	170	61.1	31	15.6	35	13.1	31	10.2	
Education level												
	<High School	242	6.9	158	64.2	41	21.4	24	8.2	19	6.2	0.060
	High School	323	11.1	201	62.1	46	15.6	54	13.2	22	9.1	
	>High School	1719	80.7	1222	71.1	175	10.5	178	10.5	144	7.9	
Region												
	Northeast	327	15.7	217	64.6	38	12.8	31	10.4	41	12.2	0.013
	Mid-west	440	19.7	322	74.9	40	9.9	41	6.8	37	8.5	
	South	1046	43.1	694	66.1	131	12.4	140	13.8	81	7.7	
	West	511	21.5	377	75.4	59	11.9	46	8.1	29	4.7	
Employment												
	Employed	666	33.7	500	75.5	55	8.8	65	8.8	46	6.9	0.021
	Not employed	1658	66.3	1110	66.6	213	13.4	193	11.6	142	8.4	
Poverty status												
	Poor	580	16.7	361	61.7	89	15.6	75	12.9	55	9.9	0.022
	Near Poor	592	23.1	397	68.3	72	12.5	74	10.6	49	8.6	
	Middle Income	560	25.6	387	66.9	56	12.0	71	13.3	46	7.8	
	High Income	592	34.6	465	76.3	51	9.5	38	7.7	38	6.6	
Health Insurance												
	Private	1013	52.1	760	74.3	88	9.3	98	9.6	67	6.7	0.013
	Public	1271	46.3	822	64.4	177	14.8	155	11.7	117	9.1	
	Uninsured	40	1.6	28	68.3	3	6.6	5	15.1	4	9.9	
Rx Insurance												
	Rx insurance	729	38.9	545	74.1	64	9.6	67	8.9	53	7.4	0.099
	No Rx insurance	1595	61.1	1065	66.7	204	13.2	191	11.8	135	8.2	
General health												
	Excellent/very good	501	24.4	392	75.7	49	10.7	45	11.0	15	2.6	<0.0001
	Good	799	34.7	592	74.2	76	10.0	89	10.9	42	4.9	
	Fair/poor	1024	40.9	626	62.0	143	14.1	124	10.3	131	13.6	
Physical activity												
	3/week	906	40.6	636	69.7	89	8.8	119	14.0	62	7.4	0.001
	No exercise	1412	59.2	968	69.4	179	13.9	139	8.4	126	8.3	
Heart												
	Yes	446	20.1	291	65.5	58	13.0	53	12.4	44	9.0	0.591
	No	1878	79.9	1319	70.6	210	11.5	205	10.2	144	7.6	
Hypertension												
	Yes	1355	54.3	913	66.8	170	13.5	163	11.6	109	8.1	0.122
	No	969	45.7	697	72.9	98	9.8	95	9.6	79	7.7	
Diabetes												
	Yes	583	21.6	402	71.9	75	11.8	52	8.1	54	8.1	0.424
	No	1741	78.4	1208	69.0	193	11.8	206	11.4	134	7.9	
Hyperlipidemia												
	Yes	1028	43.7	217	64.2	47	15.1	34	9.8	37	10.9	0.178
	No	1296	56.3	1393	70.6	221	11.2	224	10.8	151	7.4	
Asthma												
	Yes	410	17.1	248	59.8	57	15.3	48	10.4	57	14.5	0.001
	No	1914	82.9	1362	71.6	211	11.1	210	10.7	131	6.5	
COPD												
	Yes	277	10.7	154	56.5	36	13.2	43	13.0	44	17.3	0.000
	No	2047	89.3	1456	71.2	232	11.7	215	10.4	144	6.8	
Osteoarthritis												
	Yes	335	15.3	679	65.5	128	13.2	142	14.2	79	7.1	0.002
	No	1989	84.7	931	72.8	140	10.8	116	7.9	109	8.5	
GERD												
	Yes	499	20.6	298	60.2	69	14.9	63	11.6	69	13.3	<0.0001
	No	1825	79.4	1312	72.0	199	11.0	195	10.4	119	6.5	
Cancer												
	Yes	238	10.0	166	66.9	29	12.5	21	11.1	22	9.5	0.863
	No	2086	90.0	1444	69.9	239	11.8	237	10.6	166	7.7	

*p* value represents baseline differences between Rheumatoid Arthritis groups from chi-square tests. COPD: Chronic obstructive pulmonary disease; Dep: Depression; GERD: Gastro Esophageal reflux disease; Wt: weighted; RA: Rheumatoid arthritis; Rx: Medication.

**Table 2 jcm-14-07940-t002:** Weighted Means and Standard Errors of HRQoL Scores by Rheumatoid Arthritis Groups.

		Total Sample		RAs Only		RA & Dep		RA & Anxiety		RA & Dep & Anxiety		
		Mean	SD	Mean	SE	Mean	SE	Mean	SE	Mean	SE	*p*-Value
HRQoL												
	PCS	36.16	12.23	38.53	0.46	35.50	1.05	36.23	1.21	33.99	1.26	<0.0001
	MCS	47.56	11.28	51.15	0.33	42.74	1.05	43.85	0.92	38.55	1.01	<0.0001

Asterisks represent significant mean differences by Rheumatoid Arthritis groups using Anova. Dep: Depression; HRQoL: Health-related Quality of Life; MCS: Mental Component Summary; PCS: Physical Component Summary; SE: Standard Error; SD: Standard Deviation.

**Table 3 jcm-14-07940-t003:** Factors Related to Physical and Mental HRQoL among Adults with Rheumatoid Arthritis from Adjusted Multivariate Linear Regressions Analysis.

		Physical Component Summary HRQoL			Mental Component Summary HRQoL		
		B (Unstd.)	SE	Sig.	B (Unstd.)	SE	Sig.
Hyperlipidemia Group							
	RA & depression	−0.450	0.055	***	−6.608	0.106	***
	RA & anxiety	−1.101	0.131	***	−6.836	0.015	***
	RA & depression & anxiety	0.071	0.143		−9.766	0.116	***
	RA only (Ref.)						
Age in years							
	22–39	2.747	0.066	***	−2.462	0.046	***
	40–49	0.598	0.051	***	−2.148	0.025	***
	50–64	0.419	0.060	***	−2.767	0.029	***
	>65 (Ref.)						
Gender							
	Men	0.203	0.087	*	0.088	0.024	***
	Women (Ref.)						
Race/ethnicity							
	African American	0.494	0.037	***	0.550	0.038	***
	Latino	2.055	0.052	***	−0.666	0.034	***
	Others	1.930	0.007	***	−2.039	0.015	***
	White (Ref.)						
Marital status							
	Married	0.511	0.027	***	1.015	0.016	***
	Widow/Separated/Divorce	−0.209	0.042	***	1.177	0.011	***
	Never married (Ref.)						
Education level							
	>High School	−0.421	0.074	***	1.160	0.084	***
	High School	−0.066	0.063		−0.153	0.061	*
	<High School (Ref.)						
Region							
	Northeast	0.256	0.008	***	−0.475	0.012	***
	Mid-west	2.138	0.019	***	0.375	0.024	***
	South	−0.965	0.059	***	−0.173	0.008	***
	West (Ref.)						
Employment							
	Employed	4.439	0.042	***	0.941	0.046	***
	Not employed (Ref.)						
Poverty status							
	Poor	−2.281	0.036	***	−2.612	0.051	***
	Near Poor	−3.570	0.046	***	−2.169	0.028	***
	Middle Income	−2.218	0.033	***	−1.179	0.047	***
	High Income (Ref.)						
Medication Insurance							
	Medication insurance	−1.322	0.009	***	−2.139	0.038	***
	No Medication insurance (Ref.)						
General health							
	Ex/very good	12.315	0.022	***	5.118	0.026	***
	Good	6.910	0.067	***	4.784	0.059	***
	Fair/poor (Ref.)						
Physical activity							
	3 times per week	3.436	0.016	***	1.252	0.041	***
	No exercise (Ref.)						
Heart							
	Yes	−0.968	0.048	***	−0.198	0.077	*
Hypertension							
	Yes	−1.274	0.060	***	0.187	0.024	***
Diabetes							
	Yes	−1.584	0.022	***	−0.599	0.009	***
Asthma							
	Yes	−1.199	0.015	***	−1.487	0.036	***
COPD							
	Yes	−1.430	0.072	***	1.350	0.068	***
GERD							
	Yes	−0.288	0.007	***	−0.398	0.069	***
Cancer							
	Yes	−0.100	0.109		−0.130	0.010	***

Asterisks denote statistical significance in parameter estimates from multivariate linear regressions on health-related quality of life. *** *p* < 0.001; * 0.01 ≤ *p* < 0.05. COPD: Chronic obstructive pulmonary disease; GERD: Gastroesophageal reflux disease; RA: Rheumatoid arthritis; Ref: reference group; SE: Standard Error; Sig: Significance. B (Unstd.): Unstandardized regression coefficient.

## Data Availability

The dataset used in this study is available from the MEPS database at this URL: https://meps.ahrq.gov/mepsweb/data_stats/download_data_files.jsp (accessed on 15 September 2025).

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
