# Peer review of "Depression, Anxiety, and Health-Related Quality of Life in Adults with Rheumatoid Arthritis: Findings from a National Survey"

_jcm, 2025, doi:10.3390/jcm14227940_

Round 1
Reviewer 1 Report
Comments and Suggestions for Authors
I want to thank the Author and the Editors for the opportunity to review the article submitted to MDPI’s Journal of Clinical Medicine. The reviewed manuscript refers to the ill-being and HRQoL among individuals suffering from rheumatoid arthritis (RA). The presented introduction is sufficient, but unfortunately the results section needs a significant revisions before additional evaluation. Below you will find my suggestions for some minor and major changes.
Minor changes:
Introduction: While this section gives a proper explanation of the examined problem, it is solely based on the results of scientific research. I highly encourage the author to supplement this section with information from the current diagnostic criteria of ICD-11 on rheumatoid arthritis (code FA20 in ICD-11).
Results: the authors is reporting the unstandardised OLS regression coefficient B (large b) as Beta (β), for example on page 6 we can see “β =4.44; p<0,001”. Standardised Beta coefficients have a range of -1 to +1. The author is reporting the unstandardised coefficients. Please correct the used symbols so it is not misleading.
Major changes:
Results: the authos interprets their results solely based on p-values, without calculating effect size measures. P-values are highly correlated with size of the sample. The presented results are based on a sample of N=2324. On such large sample size, even small effect sizes can be significant. Effect size measures are free from sample size bias. It is crucial to report them, because there is a difference between stating that:
- Two groups of patients are significantly different vs. two groups are different but the difference is very small, or
- That there is a significant relationship between age and HRQoL vs. stating that this relationship is significant but weak/moderate/strong.
I’ve included a table with a suggestion on which effect size measures could be reported for analyses used in the author’s manuscript.
Results: the presented results lack proper test-statistics. Reporting all test statistics is crucial so the reader can evaluate if the analysed results are free from any p-hacking practices (see Table below). If the authors does not want to include them in the manuscripts’ body due to space limitations, I highly suggest that they are present as a supplementary file or via an open access archive such as OSF.
|
Analysis |
Effect size measure |
Test statistics |
|
Chi-squared |
Yule’s Phi (for 2x2 tables) and Cramer’s V (for larger tables) |
Chi-squared; degrees of freedom; p-values / confidence intervals |
|
ANOVA |
Partial eta-squared or omega-squared (less biased) |
F-statistic, df1, df2, p-value |
|
Tukey’s HSD post-hoc |
Cohen’s d |
Tukey’s t-statistic; df; p-value |
|
OLS regression |
Cohen’s f-squared (calculated from R-squared) |
Standardised coefficient Beta (β), unstandardised coefficient (B), standard errors (SE), p-values; R-squared values |
Discussion: this section should be rewrittne based on the new Results section obtained after calculating effect size measures.
In summary, I believe that the manuscript has a very large potential, but the author needs to report effect size measures for all performed analyses, and then revise the results section and discussion section accordingly based on the obtained effect sizes. With those changes in mind, the manuscript can be evaluated once again.
I hope that the Author find my suggestions valuable.
Author Response
We sincerely thank the Editor and Reviewers for their valuable time, insightful comments, and constructive feedback, which have substantially enhanced the clarity, rigor, and overall quality of our manuscript. All corresponding revisions have been incorporated into the manuscript and are highlighted accordingly, with Tables Effect Size provided in Appendix A.
Reviewer 1
Minor Changes
Comment: Introduction: While this section gives a proper explanation of the examined problem, it is solely based on the results of scientific research. I highly encourage the author to supplement this section with information from the current diagnostic criteria of ICD-11 on rheumatoid arthritis (code FA20 in ICD-11).
Response: We appreciate the reviewer’s valuable suggestion. We have added to the introduction section the current diagnostic criteria of ICD-11 on RA and mentioned in the methods that current MEPS dataset utilizes ICD-10-CM codes.
Comment: Results: the author is reporting the unstandardised OLS regression coefficient B (large b) as Beta (β). For example, on page 6 we can see “β = 4.44; p < 0.001”. Standardised Beta coefficients have a range of −1 to +1. The author is reporting the unstandardised coefficients. Please correct the used symbols so it is not misleading.
Response: We thank the reviewer for noting this error. The manuscript has been revised to correctly report unstandardised coefficients as B instead of β. Table 3 have been updated accordingly.
Major changes:
Comment: Results: The author interprets their results solely based on p-values, without calculating effect size measures. P-values are highly correlated with sample size (N = 2324). Effect sizes are essential to interpret the practical significance of the results.
Response: We fully agree. Following this recommendation, we have now computed and reported appropriate effect size measures for all analyses. These have been incorporated into the Results section and summarized in Appendix A.
Comment: Results: the presented results lack proper test statistics. Reporting all test statistics is crucial so the reader can evaluate the analyses. If space is limited, these can be provided as supplementary material or via open access (e.g., OSF).
|
Analysis |
Effect size measure |
Test statistics |
|
Chi-squared |
Yule’s Phi (for 2x2 tables) and Cramer’s V (for larger tables) |
Chi-squared; degrees of freedom; p-values / confidence intervals |
|
ANOVA |
Partial eta-squared or omega-squared (less biased) |
F-statistic, df1, df2, p-value |
|
Tukey’s HSD post-hoc |
Cohen’s d |
Tukey’s t-statistic; df; p-value |
|
OLS regression |
Cohen’s f-squared (calculated from R-squared) |
Standardised coefficient Beta (β), unstandardised coefficient (B), standard errors (SE), p-values; R-squared values |
Response: We appreciate this excellent suggestion. The revised manuscript now includes these test statistics in the Supplementary Appendix A.
Comment: Discussion: This section should be rewritten based on the new results section obtained after calculating effect size measures.
Response: We have revised the Discussion section accordingly. The updated text now integrates interpretations based on effect size magnitudes, distinguishing between statistically significant but small effects and more meaningful clinical differences.
Reviewer 2 Report
Comments and Suggestions for Authors
I appreciated having the opportunity to review this manuscript that analyzes data from the U.S. Medical Expenditure Panel Survey (2017–2022) to assess how comorbid depression and anxiety affect health-related quality of life (HRQoL) in adults with rheumatoid arthritis (RA).
I just have some comments that I think can help improve it.
Please clarify how depression and anxiety were identified (was it self-reported vs. physician-diagnosed?
In my opinion it would be better to address disease characteristics, please state RA duration, patients treatment status, and activity measures
Please note that the results and discussion sections restate the same associations repeatedly. Focus on what is new to this article.
Consider adding a figure (forest plot of regression coefficients) to illustrate the relative impact of depression and anxiety on HRQoL.
Author Response
We sincerely thank the Editor and Reviewers for their valuable time, insightful comments, and constructive feedback, which have substantially enhanced the clarity, rigor, and overall quality of our manuscript. All corresponding revisions have been incorporated into the manuscript and are highlighted accordingly, with Tables Effect Size provided in Appendix A.
Reviewer 2
Comment: Please clarify how depression and anxiety were identified (was it self-reported vs. physician-diagnosed?)
Response: Thank you for this important clarification request. Depression and anxiety were self-reported by participants during structured household interviews and subsequently coded into standardized International Classification of Diseases, Tenth Revision, Clinical Modification (ICD-10-CM) codes by trained professional coders employed by the Agency for Healthcare Research and Quality (AHRQ). We have now clarified this information in the Methods section of the revised manuscript.
Comment: In my opinion, it would be better to address disease characteristics; please state RA duration, patients’ treatment status, and activity measures.
Response: We appreciate this insightful comment. Unfortunately, MEPS dataset does not contain variables capturing rheumatoid arthritis (RA) duration, treatment regimen, or disease activity measures. We agree that these characteristics are important determinants of HRQoL. Accordingly, this limitation has now been explicitly acknowledged in the discussion section, noting that the absence of such clinical details may constrain the interpretation of the findings.
Comment: Please note that the results and discussion sections restate the same associations repeatedly. Focus on what is new to this article.
Response: Thank you for this constructive feedback. We have streamlined both the Results and Discussion sections to minimize redundancy. The revised text now emphasizes the novel contributions of this study, particularly the comparative assessment of HRQoL across comorbid RA-depression-anxiety subgroups and the integration of effect size measures to quantify clinical significance beyond p-values.
Comment: Consider adding a figure (forest plot of regression coefficients) to illustrate the relative impact of depression and anxiety on HRQoL.
Response: We appreciate this excellent suggestion. A forest plot summarizing the standardized (β) and unstandardized regression coefficients for both Physical (PCS) and Mental (MCS) Component Summary scores has now been added as Figure 1 in the revised manuscript. This visual representation enhances the clarity and interpretability of the relative effects of depression and anxiety on HRQoL outcomes.
Reviewer 3 Report
Comments and Suggestions for Authors
The author examined the association between RA patients with different mental health conditions and their HRQoL scores. The study was well designed and thorough, with only minor suggestions for additional analysis to further strengthen the manuscript.
- It would be interesting to also examine the impact on HRQoL scores by mental health conditions in healthy individuals and determine whether mental health conditions and RA has a simple additive or a compouding effect on HRQoL.
Author Response
We sincerely thank the Editor and Reviewers for their valuable time, insightful comments, and constructive feedback, which have substantially enhanced the clarity, rigor, and overall quality of our manuscript.
Reviewer 3
Comment: It would be interesting to also examine the impact on HRQoL scores by mental health conditions in healthy individuals and determine whether mental health conditions and RA have a simple additive or a compounding effect on HRQoL.
Response: We appreciate this insightful suggestion. Unfortunately, the current study focused exclusively on individuals with a diagnosis of rheumatoid arthritis (RA) as identified in the MEPS dataset. As such, comparable data for “healthy individuals” without RA were not analyzed within this framework. We agree that comparing the impact of mental health conditions on HRQoL between individuals with and without RA would provide valuable insights into potential additive versus compounding effects. We have now acknowledged this as a future research direction in the Discussion section, highlighting the importance of extending this work to population-based comparisons that include non-RA groups.
Round 2
Reviewer 1 Report
Comments and Suggestions for Authors
I would like to sincerely thank the author for the opportunity to review the revised version of their manuscript. I am pleased that my comments were helpful. The author addressed all my concerns. I consider their responses and the changes they implemented as fully justified. I have no additional comments about the reviewed article.
Author Response
We sincerely thank the reviewer for their thoughtful feedback and careful evaluation of our revised manuscript. We are grateful for the time and effort devoted to reviewing our work and for the constructive comments that helped us improve the clarity and rigor of the paper. We are pleased that the revisions have satisfactorily addressed all concerns, and we deeply appreciate the reviewer’s positive assessment and support of our study.
Reviewer 2 Report
Comments and Suggestions for Authors
the authors replied and modified the manuscript accordingly to my suggestions.
Author Response
We sincerely thank the reviewer for their constructive feedback and kind remarks. We are truly grateful for the time and expertise invested in reviewing our manuscript and for recognizing our efforts in addressing the previous comments. Your thoughtful guidance greatly contributed to enhancing the quality and clarity of our work. We deeply appreciate your positive evaluation and encouraging words.